# Development of Anti-LRRC15 Small Fragments for Imaging Purposes Using a Phage-Display ScFv Approach

**DOI:** 10.3390/ijms232012677

**Published:** 2022-10-21

**Authors:** Pierre-Emmanuel Baurand, Jérémy Balland, Chloé Reynas, Mélanie Ramseyer, Delphine Vivier, Pierre-Simon Bellaye, Bertrand Collin, Catherine Paul, Franck Denat, Kamal Asgarov, Jean-René Pallandre, Laurence Ringenbach

**Affiliations:** 1Diaclone SAS-Part of Medix Biochemica Group, 6 Rue Dr Jean-François-Xavier Girod, BP 1985, 25000 Besançon, France; 2Institut de Chimie Moléculaire de l’Université de Bourgogne, UMR CNRS 6302, Université de Bourgogne Franche-Comté, 21000 Dijon, France; 3Plateforme D’imagerie et de Radiothérapie Précliniques (PIRP), Service de Médecine Nucléaire, Centre Georges-François Leclerc, 1 Rue du Pr Marion, 21000 Dijon, France; 4Laboratoire d’Immunologie et Immunothérapie des Cancers, EPHE, PSL Research University, 75000 Paris, France; 5LIIC, EA7269, Université de Bourgogne Franche Comté, 21000 Dijon, France; 6INSERM, EFS BFC, UMR1098, RIGHT, Interactions Greffon-Hôte-Tumeur/Ingénierie Cellulaire et Génique, University of Bourgogne Franche-Comté, 25000 Besançon, France; 7Clinical Investigation Center in Biotherapy, INSERM CIC-BT1431, Besançon University Hospital, 25000 Besançon, France; 8ITAC Platform, University of Bourgogne Franche-Comté, 25000 Besançon, France

**Keywords:** phage display, ScFv immune libraries, imaging agents, LRRC15, Cys-ScFv, Cys-diabodies, immunotherapy, small antibody fragments

## Abstract

The human leucine-rich repeat-containing protein 15 (LRRC15) is a membrane protein identified as a marker of CAF (cancer-associated fibroblast) cells whose overexpression is positively correlated with cancer grade and outcome. Nuclear molecular imaging (i.e., SPECT and PET) to track LRRC15 expression could be very useful in guiding further therapeutic strategies. In this study, we developed an ScFv mouse phage-display library to obtain small fragment antibodies against human LRRC15 for molecular imaging purposes. Mice were immunized with recombinant human LRRC15 (hLRRC15), and lymph node cells were harvested for ScFv (single-chain variable fragment) phage-display analysis. The built library was used for panning on cell lines with constitutive or induced expression after transfection. The choice of best candidates was performed by screening various other cell lines, using flow cytometry. The selected candidates were reformatted into Cys-ScFv or Cys-diabody by addition of cysteine, and cloned in mammalian expression vectors to obtain batches of small fragments that were further used in site-specific radiolabeling tests. The obtained library was 1.2 × 10^7^ cfu/µg with an insertion rate >95%. The two panning rounds performed on cells permittedenrichment of 2 × 10^−3^. Screening with flow cytometry allowed us to identify 28 specific hLRRC15 candidates. Among these, two also recognized murine LRCC15 and were reformatted into Cys-ScFv and Cys-diabody. They were expressed transiently in a mammalian system to obtain 1.0 to 4.5 mg of Cys fragments ready for bioconjugation and radiolabeling. Thus, in this paper, we demonstrate the relevance of the phage-display ScFv library approach for the fast-track development of small antibodies for imaging and/or immunotherapy purposes.

## 1. Introduction

Cancer represents one of the main causes of morbidity and mortality worldwide, with about 19.3 million new cases and 10 million cancer-related deaths in 2020. Approximately one man in five and one woman in six will have cancer during their lifetime [1]. Research has developed a therapeutic arsenal, and for the past thirty years a revolution has been achieved in “targeted immunotherapy”, including immune-checkpoint inhibitors [2]. To date, the three primary targets of checkpoint inhibition include the programmed death protein-1 receptor (PD-1), its ligand (programmed-death ligand-1 or PD-L1), and the cytotoxic T-lymphocyte-associated antigen-4 receptor (CTLA-4). From these targets, several antibody-based checkpoint inhibitors have been approved for clinical use, and many others are still in clinical trials [3]. Despite the undeniable contribution of immunotherapy to cancer treatment, this type of treatment is not effective in all patients, since fewer than 50% of patients respond to these immunotherapies [4], highlighting the great need for surrogate biomarkers of response to immunotherapy. The choice and use of biomarkers remain crucial for patients. One of the new challenges in the field of immunotherapy is to identify accurate and reproducible biomarkers that allow treating physicians to select the treatments to which patients are most likely to respond. In terms of responses to anti-PD-1/PD-L1, biomarkers such as tumor mutational load, PD-L1 expression, and intra-tumoral cytotoxic CD8+ T-cell infiltrates have been proposed, but further investigations are necessary to select the most suitable [5]. Current methods for monitoring T cells (blood, biopsies) [6] do not necessarily reflect the dynamic and spatial information needed to assess the immune response to therapeutic interventions. Thus, specific in vivo molecular imaging (e.g., positron emission tomography—PET) of these cytotoxic CD8+ T cells could circumvent these drawbacks, with the capability of noninvasively monitoring systemic and intra-tumoral alterations in the numbers or localization of immune cells [7,8]. Despite its high sensitivity and specificity, PET is limited by low spatial resolution and the inability to provide anatomical detail, which could be balanced using a hybrid integrated PET/MRI scan.

LRRC15 (also known as hLib), a type I transmembrane protein and member of the leucine-rich repeat superfamily [9,10], is frequently overexpressed in various tumor types, such as prostate, breast, ovarian, and cervical tumors [11,12], and can also cause resistance to adenoviral p53 [13]. Recent studies have highlighted LRRC15 expression targeting mesenchymal cells [14] and cancer-associated fibroblasts (CAF) [15]. The LRRC15+ CAF signature correlates with poor response to immune-checkpoint blockade in several different human tumor types [15,16]. In 2021, Ray U et al. [17] determined a mechanistic link between LRRC15 expression and the promotion of ovarian cancer metastasis. In the past three years, several studies targeting LRRC15 using antibody–drug conjugates (ADC) have shown promising results [18,19,20,21].

The use of nuclear molecular imaging (i.e., single photon emission computed tomography—SPECT and PET) to track LRRC15 expression could potentially guide further therapeutic strategies, as cell-surface markers are now commonly used as targets for the development of therapeutic or imaging agents.

Over the past three decades, monoclonal antibodies have become the market’s most important class of therapeutic biologicals. Monoclonal antibodies were initially produced by the hybridoma method, in use since 1975 [22], but emerging recombinant DNA technologies have allowed the humanization of murine monoclonal antibodies and made them suitable for the treatment and diagnosis of chronic conditions including cancer and autoimmune diseases [23]. In addition, molecular engineering approaches, including phage-display technology, enable the expression of antibody libraries on filamentous phages to generate antigen-specific candidates [24]. Phage display enables the production of small recombinant antibody fragments such as Fab or ScFvs [25]. ScFvs are non-natural small antibody formats (~25 kDa, monomeric) resulting from genetic engineering, and are well-expressed in bacterial and phage systems. Phage immune libraries, notably ScFv, have demonstrated their power for discovering high-affinity binders against various types of targets [25].

Dimeric ScFvs, called diabodies (~50 kDa), are engineered antibody fragments that have already been demonstrated by several teams as useful for developing agents in tumor-imaging applications [24,25,26,27], and include the monomeric ScFv [28]. In these previous works [24,25,26,27], imaging agents were taken from full antibody candidates against HER2 and CD20, and were reformatted as Cys-diabodies (Cys-Db). These engineered small dimeric fragments were subjected to site-specific conjugation followed by radiolabeling, and were successfully used as tumor tracers in PET applications [24,26]. These studies demonstrated that Cys-Db approaches could be applied to antibodies against cell surface biomarkers.

To our knowledge, and surprisingly, no research has yet used phage-display technology, especially the ScFv immune libraries approach, to generate small antibody fragments usable as imaging agents. ScFv phage-display libraries offer the huge advantage of directly selecting candidates with a format close to the desired final type (in our case, Cys-ScFv or Cys-diabody). This approach appears very promising for discovering new small antibody fragments for use as trackers for in vivo molecular imaging.

Our study aims to demonstrate that the phage-display approach using the ScFv immune library is adaptable for the fast discovery of small new antibody fragments usable as imaging agents.

## 2. Results

### 2.1. Immunization

Four days after the last injection of LRRC15-hFc recombinant antigen and one day before the cells’ collection, sera specificity of six hLRRC15 foot-pad immunized mice were tested by indirect flow cytometry (FC) surface staining on HeLa LRRC15 and U87-MG cells, respectively, expressing exogenous and endogenous hLRRC15. An immune response was observed even at the highest dilution conditions tested (1/10,000). As shown in Figure 1, a high level of immunization was obtained with very good reproducibility (standard deviation ≤ 5%) between the six sera. The results obtained for each mouse are given in Appendix A.

**Figure 1 ijms-23-12677-f001:**
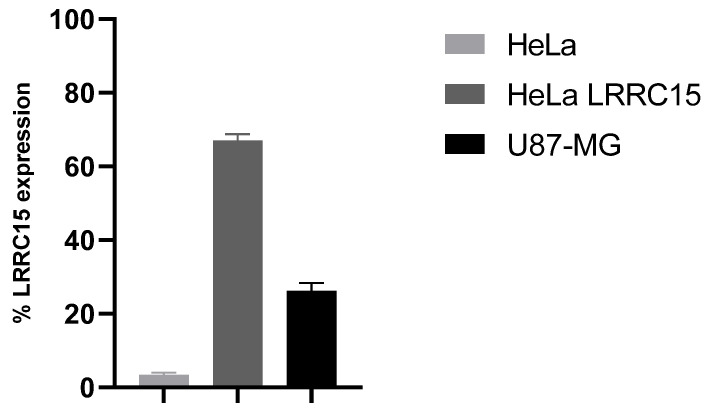
Sera LRRC15 specificity test. Sera of six mice were tested (1000 times diluted) on cells with no hLRRC15 expression (HeLa), recombinant hLRRC15 cells (HeLa hLRRC15), and cells constitutively expressing hLRRC15 (U87-MG, without TGFBeta induction). Mean value of LRRC15 labeling is given in %. The lymph nodes of each mouse were harvested, and immune cells were extracted by enzymatic digestion and then counted. The same quantities of 30 × 10^6^ cells per mouse were picked and pooled. The mRNA extraction was completed on 180 × 10^6^ cells from the pool (Table 1).

### 2.2. Library Construction

The quality of the ScFv fragments obtained after three rounds of PCR was assessed by agarose 1.2% (*w*/*v*) gel electrophoresis in a TAE 1X buffer (30 min at 135 V), after the purification step with agarose following the kit protocol for Nucleospin PCR and Clean-up (Macherey-Nagel, Oensingen, Germany). As shown in Figure 1, clear ScFv VH and VL fragments near 500 and 400 bp were obtained, respectively. These two pools of ScFv fragments were used in recombination reactions to obtain a full ScFv final insert (VH + VL). After purification, a clear and proper 900 bp fragment (without visible contaminant bands < 900 bp) was visualized on agarose gel 1.2% (*w*/*v*; TAE 1X) (Figure 2).

The quality of the ScFv library was evaluated with three criteria:-The size, assessed by spotting 5 µL of 10 times serial dilution of TG1 cells after electroporation. For the given testing conditions, an acceptable size was >10^6^ colony-forming unit (cfu/µg).-The insertion rate (IR), determined after colony PCR on randomly chosen clones from the bacterial spread of the electroporated library. A clone was deemed positive when the PCR indicated a 1100 bp fragment. IR was assessed as acceptable when it included more than 80% of positive clones.-The quality of PCR inserts cloned in our phagemid (expression vector usable for phage display) was also assessed by the sequencing of 96 randomly chosen positive PCR clones, to determine the % of coding ScFv sequences showing an ORF. In our study, the number of sequences presenting an ScFv ORF was >90%, which was higher than our own criteria of >80% (Table 2). To validate the library, the global diversity was confirmed by a low level of identical sequences (less than 10%). In this study, no redundance was found for the analyzed clones. Summaries of clone sequences and diversity analyses are presented in Appendix A.

As shown in Table 2, high percentages of obtained results demonstrated the accuracy of the procedures performed.

The obtained library size was 1.2 × 10^7^ cfu/µg, with a very high level of insertion (>95% of positive inserts, 151/156 clones). A total of 88 sequences among the 96 analyzed clones showed correct ORF encoding complete ScFvs, without redundance between the 88 coding clones. The diversity of the sampling was maximal.

### 2.3. Panning—Selection

The selection results obtained were very satisfactory, with ratios of retention of 3 × 10^−^^7^ after the first panning round (HeLa hLRRC15 cells) and 2 × 10^−^^3^ after the second panning round on NHI3T3 hLRRC15 cells (Table 2). The enrichment factor was 1000 (10^−^^7^ to ^−^^3^); this factor was higher than our standard of 100, which demonstrated that no further panning rounds were necessary.

### 2.4. Screening by Phage Flow Cytometry

After the selection of the ScFv library over two panning rounds, the next step was to identify ScFv clones recognizing the human native form and also the murine recombinant form of the LRRC15 marker. For this purpose, several cell-screening steps were performed by flow cytometry on ScFv periplasmic extracts.

Firstly, to select clones targeting hLRRC15, screening was performed on recombinant human LRRC15 transfected HeLa and NIH3T3 cell lines. More than 80% of the clones recognized the hLRRC15 recombinant form on HeLa and NIH3T3 (Table 3).

Secondly, after the first sort, a new screen was performed on U87-MG cells to select clones also targeting the hLRRC15 native form. The U87-MG cells were cultured 24 h before the screen, in the presence of TGF beta to induce a high expression level of LRRC15 marker on the cell surface. This step allowed us to identify 28 clones from among the hLRRC15 recombinant-specific candidates that also recognized the hLRRC15 native form (Table 3). Identified positive clones were sequenced. A total of 17 and 24 different CDR3 domains were identified for the VH and VL domains, respectively (Table 4). CDR3 sequences are given in Appendix A.

Thirdly, screening of HEK293 cells overexpressing mLRRC15 cells was performed to identify clones also specific to the mLRRC15 recombinant form. Among all clones screened, two recognized both human LRRC15 and murine recombinant LRRC15, clones F4 and B3.1 (Table 3).

### 2.5. Reformatting and Validation of Small Recombinant Fragments

After engineering by molecular biology, pilot batches of reformatted recombinant clones were produced for 14 days in 60 mL of transfected CHO cells. After production, Cys-ScFv and Cys-Db were purified. As shown in Table 5, between 1 and 4.5 mg of recombinant purified small fragments were obtained.

The quality of purified small fragments was visually assessed after electrophoresis on SDS page gel 4–15%. As shown in Figure 3, the purity was high (>85%).

The theoretical molecular weight of the F4 Cys-ScFv candidate was 27.7 kDa. The reduced form had a size close to 25 kDa, whereas the non-reduced form showed a minority of dimeric forms near 50 kDa (due to the presence of extra cysteine forming disulfide bridges) and a majority of monomeric forms near 25 kDa (Figure 3A).

The theoretical molecular weight of the B3.1 Cys-Db candidate was 26.3 kDa. The reduced form had a size near 25 kDa, whereas the non-reduced form showed a majority of dimeric forms close to 50 kDa (due to the presence of extra cysteine forming disulfuric bridges), compared to the monomeric form near 25 kDa (Figure 3B).

### 2.6. Validation of Small Engineering Recombinant Fragments

Specificity validation by FC was performed on reformatted small antibodies, to ensure they were specific to the natural and recombinant human forms and the recombinant murine form. The cell lines were checked for LRRC15 expression, as shown in Figure 4. The two versions of the two final candidates were tested on U87-MG (endogenously expressing hLRRC15), transfected NIH3T3 and HeLa cell lines (recombinant hLRRC15), and the HEK293 cell line (recombinant mLRRC15). Cytometry profiles confirmed that Cys-ScFv and Cys-Db from both clones were specific to the natural and recombinant human forms and recognized the recombinant murine form (Figure 4). F4 candidates in Cys-ScFv and Cys-Db formats showed higher labeling than B3.1 candidates (Figure 4 and Table 6). The Cys-ScFv F4 candidate showed higher labeling in the natural form (U87-MG) than the full antibody B-G53 obtained via the classical hybridoma metho, or Cys-Db F4, or the two-Cys fragment B3.1.

## 3. Discussion

The use of ScFv immune phage-display libraries has already shown good results in the development of human therapeutic recombinant antibodies for tumor-targeted therapy EphA2 [29], and in the discovery of ScFv against IL1RAP from a human library [30].

To our knowledge, no work is available in the literature reporting the use of the ScFv library in the discovery or development of imaging agents. Based on our results, the use of the ScFv library is appropriate because (1) phage-display technology only permits monovalent expression, and it is impossible to express diabodies directly on the phage, and (2) the ScFv is closer to the diabody format (with a dimeric form and a smaller link between variable domains). Our innovative approach has the main advantage of limiting the loss of target affinity and avidity, which can often occur after reformatting a full antibody into small antibody fragments. In the literature, studies have described the use of Cys small fragments from reformatted full antibodies [24,26] and have not involved ScFv phage-display technology.

For the discovery of candidates against a membrane protein (cell-surface biomarkers), Diaclone phage-display ScFv technology appears very powerful. In the present work, the fast-track development of anti-LRRC15 small antibody fragments starting from an ScFv phage-display immune library has been demonstrated.

Indeed, in approximately six months, we succeeded in production of recombinant antigen, immunization, library building, biopanning, and screening of phage ScFv candidates on various cell lines. The first batches of Cys-ScFv and Cys-Db were obtained in one month. The library building was a success, with very high good insertion (>95%) and large diversity of coding sequences. The optimized protocol developed and used by our team allowed the design of library ready to be used for the panning step in less than 10 days. ScFv construction in the phagemid was completed with VH-VL orientation using a linker length of 16 aa residues, and allowed us to select specific candidates.

After selection and screening, the two best anti-hLRRC15-specific candidates (F4 and B3.1) that also recognized mLRRC15 were engineered by the addition of extra cysteine to create Cys-ScFv (monovalent) and by shortening the linker to 5 aa to produce Cys-Db (bivalent). The F4 candidates, regardless of format, showed better specificity to LRRC15 in FC than the B3.1 clone. In addition, the Cys-ScFv F4 candidate showed a higher affinity for the natural form of LRRC15 than the B-G53 full antibody obtained by hybridoma technology. In terms of our FC results, ScFv showed better affinity than Db, but this point requires confirmation and may relate to the FC application itself. Finally, this indicates that Cys small fragment re-engineering is ready for bioconjugation after evaluation for its potential use in imaging applications.

ImmunoPET is commonly used in imaging for cancer diagnosis and the monitoring of treatment responses. The use of small fragment antibodies for this type of imaging has been well documented. Due to their small size, antibody fragments of <100 kDa have deeper penetration into tissue, shorter circulation times, and are not significantly metabolized or retained by the liver compared with intact antibodies [28]. Various formats are available (Fab, minibody, diabody, ScFv, nanobody, etc.). The most commonly used format is the ScFv, engineered by assembling VH and VL domains with a peptide linker of >12 amino acid residues, creating small-sized monovalent fragments (~26 kDa). The use of diabodies as imaging agents after radiolabeling is increasing [28]. These are derived from ScFv fragments and are designed by genetic engineering. The shortening of the peptide linker (close to five residues) that links the two ScFv variable domains forces them to self-assemble and create bivalent molecules (~55 kDa). All these data highlight the interest in ScFv and Db fragments for developing imaging agents.

The use of re-engineered small fragment antibodies has various applications. In ImmunoPET applications, they can be used as trackers after a chemical process of bioconjugation, followed by radiolabeling. The conjugation is performed with various chelators able to coordinate radiometals, such as DFO, NODA, NOTA, etc. [28], in a site-specific way, to obtain homogeneous labeled products. In such cases, chelators are clipped on disulfide bonds and cysteine residues [24,26,28]. Beause ScFv or Db do not exhibit disulfide bonds, extra cysteine can be added by bioengineering for site-specific conjugation, as was successfully carried out by Olafsen et al. [24,26]. Based on the same construction [25,26], we re-engineered our anti-LRRC15 small fragments with free cysteine residues in the C-terminal part as Cys-small antibodies ready for site-specific conjugation and radiolabeling. Reflecting the good results obtained by Olafsen et al. [24,26] with Cys small fragments, the LRRC15 candidates developed in our work appear to present good potential for testing with ImmunoPET applications. However, their efficiency as imaging agents remains to be determined.

Cys small fragments can also be interesting tools for other chemistry applications, such as coupling fluorescent molecules for immunochemistry detection or directed antibody–drug conjugates. Tests of Cys fragments such as ADC have become more interesting as more data become available concerning the use of full IgG ABBV-05 ADCs anti-LRRC15 as a therapeutic agent for osteosarcoma [20].

In conclusion, this work highlights the fact that ScFv phage-display technology helps to (1) develop high-affinity small fragments for FC applications and (2) produce different fragments of Cys antibodies (two candidates with two formats) for testing as imaging agents (after subsequent steps of bioconjugation and radiolabeling).

In addition to offering a good approach for small fragment discovery for various purposes, phage-display technology by ScFv immune libraries is an interesting technology in the field of cellular immunotherapy. One of the most promising cellular cancer treatments involves CAR-T cells, which are constructed based on ScFv candidates against antigens related to cancerous tumors. ScFv gives CAR-T the ability to recognize the antigen associated with immune cells, after which the CAR-T cells can specifically attack the tumor cells. Over the past 5 years, CAR-T cell treatments against lymphoma, myeloid leukemia, and multiple myeloma have been approved by the FDA [31]. This demonstrates the success of this potent new curative therapy. The use of the ScFv phage-display approach is suitable for ScFv candidate discovery for CAR-T-cell treatment development purposes, as the final antibody format can be used directly for the screening of candidates.

## 4. Materials and Methods

### 4.1. Human LRRC15

#### 4.1.1. Antigen for Immunization

The extracellular domain (Met1-Gly538) of human leucine-rich repeat-containing protein 15 (LRRC15, accession number NM_130830.5) was synthetized as a DNA string for cloning (Geneart, Regensburg, Germany). Then, the DNA string was cloned by recombination using an in-fusion cloning kit following the manual instructions (Takara Bio Europe, Saint-Germain-en-Laye, France) in fusion with the human IgG1 Fc domain (Pro100-Lys330) in a licensed optimized vector for a mammalian expression system. The LRRC15-hFc recombinant antigen was produced transiently in CHO cells (production and purification outsourced to RD-Biotech company, Besançon, France). After 14 days of production, the supernatant was purified by affinity chromatography. The purity of the antigen was evaluated after migration on 4–15% SDS-page gel under non-reducing and reducing conditions.

#### 4.1.2. Cell Lines

Recombinant NIH3T3 and HeLa cell lines expressing human full LRRC15 were a gift from UMR 1068 Right (EFS, Besançon, France), which is a partner of the BioCAIR project.

The U87-MG cell line was selected as a cell line constitutively expressing hLRRC15, based on bibliographic analyses.

### 4.2. Murine LRRC15

The full LRRC15 murine sequence (accession number NM_028973.2) was synthetized as a DNA string for cloning (Geneart, Regensburg, Germany). The DNA string was cloned by recombination using an in-fusion cloning kit following the manual instructions (Takara Bio Europe, Saint-Germain-en-Laye, France) with an N-terminal Histag in the pCDNA3.1 vector (Figure 1). This expression vector was applied to transfect transiently HEK293 cells by Lipofectamine P3000. Before testing the candidates, control of the transfection was performed by His-PE staining with flow cytometry (Cytoflex, Beckman Coulter France, Villepinte, France).

### 4.3. Reagents

The validation of hLRRC15 expression for wild-type and transfected cell lines was performed using mouse monoclonal antibody anti-hLRRC15 B-G53 (Diaclone SA, Besançon, France). Secondary antibody goat anti-mouse-IgG, PE-labeled (Abcam, Cambridge, UK), was used for detection of mouse antibodies. The biotinylated format of B-G53 was used in some experiments. The mLRRC15 expression in transfected HEK293 was monitored using mouse monoclonal antibody anti-6xHis-Tag, PE-labeled (Abcam, Cambridge, UK). Detection of biotinylated fragments and biotinylated B-G53 was performed using Streptavidin-PE (Interchim, Montluçon, France).

### 4.4. Immunization

Six Balb/c mice were immunized 5 times every week by footpad injection of 1 µg/mouse/50 µL of LRRC15-hFc recombinant antigen (the protocol has been established at Diaclone). The adjuvant used was Diaclone’s homemade one. A specificity test of sera by indirect membrane immunostaining (FC) was realized at the end of immunization using a cell line endogenously expressing hLRRC15 (U87-MG) and a cell line (HeLa) exogenously expressing hLRRC15. The HEL cell line was used as a negative control.

### 4.5. ScFv Library

#### 4.5.1. Construction

Lymph node cells from 6 immunized mice were collected after 7 weeks of immunization. Total RNA extraction started with a pool of all the lymph node cells (180 × 10^6^, Table 1) using an Rneasy Maxi kit following manual instructions for animal cells (Qiagen, Hilden, Germany). cDNA generation protocol was a basic system from Heckner et al. 2010 [32]. RNA extract (80 µg) was converted in cDNA with the SuperScript III Reverse Transcriptase (Invitrogen, Waltham, CA, USA). The ScFv library was built with orientation VH-VL: the variable domains were linked with 19 amino acids (GGGS)4. A histidine tag was followed by a Myctag in the C-terminal part of the construction (Figure 2).

IgG variable heavy chain (VH) and light chain (VL) sequences were amplified following 3 rounds of successive PCRs using our own set of primers developed at Diaclone (sequences not shown). The first round of PCR was performed to amplify VH or VL sequences starting from cDNA. Amplifications were performed with Phusion DNA Polymerase (Thermo Fisher Scientific, Waltham, MA, USA), following the PCR program: 94 °C, 2 min; 30 cycles of 94 °C, 30 s; 60 °C, 30 s; 72 °C, 60 s; final amplification at 72 °C, 7 min. The second and third rounds of PCR were performed successively on purified first and second PCR products, following the PCR program: 94 °C, 2 min; 15 cycles of 94 °C, 30 s; 62 °C, 30 s; 72 °C, 60 s; final amplification at 72 °C, 7 min. These PCR rounds added the ScFv linker to the VH or VL fragments. For VH-PCR, 11 mixes of forward and reverse primers were used. For VL-PCR, 7 mixes of forward and reverse primers were used. Complete ScFv (VH + link + VL) were built by recombination cloning. Purified final ScFv products were cloned in our own phagemid vector between *Sfi*I and *Not*I enzyme restriction sites. The library was finally constructed by the electroporation of the cloned phagemids into electrocompetent TG1 cells (Lucigen, Middleton, CA, USA).

#### 4.5.2. Validation

The library size was determined by titration after spotting 5 µL of the electroporated library diluted 10 to 10^6^ on LB-Agar plates. Insertion of the PCR product in the phagemid was evaluated as a percentage by colony PCR (using specific primers of our phagemid, surrounding the PCR insert) on 156 randomly chosen clones. The expected size for positive inserts was close to 1100 bp, corresponding to the complete ScFv sequence surrounde by cloning sites of the phagemid vector. To validate the high effectiveness of insertion and reading frames in the phagemid, a sequencing control was constructed with 96 independent positive clones.

#### 4.5.3. Phage Infection and preparation

Protocol was adapted from Kellner et al. 2010 [32]. The final library was cultured and super-infected by phage Helper M13K07 (Invitrogen, Waltham, CA, USA). After overnight culture in LB agar media supplemented with 2% glucose (at 37 °C, 200 rpm), the ScFv-phage library was precipitated by polyethylene glycol (PEG6000 20%/2.5 M NaCl). Purified phages were resuspended in 1 mL of sterile PBS.

### 4.6. Phage Display

#### 4.6.1. Panning-Selection

Two successive panning rounds were performed on different cell lines. Panning protocol was adapted from Russo et al. 2017 [33], working with microtiter plates. The first selection round was performed with HeLa cells expressing hLRRC15. A total of 10 μL of ScFv-phage library was incubated for 30 min at 4 °C with slow head rotation in 2 mL tubes containing 2.5 × 10^6^ HeLa cells. The remaining phages in the supernatant were incubated for 1 h 30 min with 10^5^ HeLa or 5 × 10^6^ HeLa hLRRC15 cells. The second selection round of panning was performed on the NIH3T3 cell line expressing hLRRC15. Here, 2 µL of phages from round 1 were incubated for 1 h with 5 × 10^6^ NIH3T3 or 5 × 10^6^ NIH3T3 hLRRC15 cells. After each round of biopanning, a new infection was performed, and phages were incubated overnight. Spots of 5 µL were placed on LB agar plates to determine the enrichment of the library between the input (before panning) and the output (after panning) in pfu/mL (phage-forming unit). Theoutput/input ratio was calculated.

#### 4.6.2. Screening

The screening method was adapted from Schirrmann, 2010 [34]. Samples were acquired on a Cytoflex (Beckman Coulter France, Villepinte, France) and analyzed using CytExpert software. Ninety randomly picked ScFv clones from the 2nd round of panning were cultured overnight in 1 mL well plates after IPTG addition to induce periplasmic expression. Periplasmic extracts containing ScFv clones were screened by flow cytometry against endogenous hLRRC15 (U87-MG cells activated with TGF beta) and against the recombinant hLRRC15 form (on HeLa and NIH3T3 cells) using a mouse anti-polyhistidine-peroxidase antibody (A7058, Sigma Aldrich, Burlington, MA, USA). Finally, to retain candidates recognizing the murine LRRC15 (mLRRC15) form, specific candidates of hLRRC15 were tested on HEK293 cells overexpressing the recombinant mLRRC15.

### 4.7. Engineering of Candidates

#### 4.7.1. Design and Cloning

The best candidates were analyzed by sequencing to identify nucleotide and amino acid sequences and eliminate redundant clones. Based on CDR3 VH and VL-domain analyses, candidates were classified into various families. For use in the subsequent steps, the 2 retained candidates were reformatted in Cys-ScFv or Cys-diabody (Cys-Db) by the addition of an extra cysteine in the C-terminal part, following the same procedure described by Olafsen et al., 2012 [24] (Figure 3).

The Cys-mab fragments were subcloned in a licenced mammalian expression vector, to produce the first batch by transient transfection in CHO cells (production and purification outsourced to RD-Biotech company, Besançon, France).

#### 4.7.2. Validation of Final Products

After purification by affinity chromatography, reformatted candidates were subjected to new validation on cell lines expressing hLRRC15 or mLRRC15 (HEK293), endogenously (U87-MG) or exogenously (HeLa and NIH3T3). Cell characterization was investigated by surface staining with previously biotinylated anti-LRRC15 small antibodies. Then, Streptavidin-PE was incubated for 30 min at 4 °C. Samples were directly acquired on a Cytoflex (Beckman Coulter France, Villepinte, France) and analyzed using CytExpert software.

## Data Availability

Not applicable.

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
