# Peer review of "Development of Anti-LRRC15 Small Fragments for Imaging Purposes Using a Phage-Display ScFv Approach"

_ijms, 2022, doi:10.3390/ijms232012677_

Round 1

Reviewer 1 Report

The paper presents the phage-display ScFv approach for development of small antibodies against the Leucine-rich repeat-containing protein 15 (LRCC15) that is expressed by cells in the cancer microenvironment. The study is relevant in perspective of developing a specific agent for in vivo imaging evaluation of cancer development and for targeted therapy. 

This original research perfectly fits with the topic of the journal.

Comments:

The manuscript is clearly organized, well documented and well written.

Introduction

The relevance to identify targeting agents in cancer against the particular antigen could be shortened. The introduction should maintain the focus on the method development of a small antibody and format derivatives as labeling markers.

Results

Immunization of mice (data in Fig 1)

The polyclonal mouse sera react with around 70% of HeLa cells expressing rhLRCC15. Please clarify, whether the reactivity corresponds to the total number of transfected cells or were only 70% of all transfected HeLa cells labeled.

In the same line, the sera show around 30% reactivity with the U87-MG cells. Please indicate reactivity in distinct culture conditions (+ or – TGFbeta). 

U87-MG cells express constitutively the transmembrane LRCC15 antigen (potentially enhanced by TGFbeta). The discrepancy of 30% positively labeled cells raises the question of the presence the antigen in the tumor cell line or rather the accessibility of the antigen to the antibody. This questions furthermore the efficacy for the attempted targeting, as the recombinant antibody fragments might even be less efficient. Indeed at the end, 2 clones were retained based on binding to U87-TG cells.

The following procedures mRNA, cDNA constructs, PCR fragments -400/500bp- and 900bp for purified ScFv and insertion are well described and documented by figures, to create the ScFv mouse phage-display library for selecting small fragment antibodies.

Table 2 / sequencing control

Please reformulate “a good open reading frame “ like  “showed correct open reading frames encoding complete ScFv” . Please present the unique obtained sequences in a Suppl File. Among these ORFs, were the same sequences of the encoding clones also found in the isolated and finally retained clones after biopanning steps? Please inform comparatively about the CDR3 domains that were identified, respectively, for the VH and VL domains.

A number of candidate clones targeting the hLRRC15 were tested with various cell lines that express the LRCC15 antigen (r human or r mouse) or the constitutive human LRCC15.  Are the positive candidates the same ones in the respective testings with these various cell lines? Please provide a Venn diagram for easier visualization of reactive clones. Why are the candidate numbers (90/80/28) different in the testings? 

The reformatted clones were tested against various cell lines expressing the LRRC15 in comparison to an available mouse monoclonal antibody. As retained being strong binders to the antigen expressed constitutively in U87-MG (under TGFbeta), clones B3.1 and in particular F4 at Cys-ScFv and Cys-Db formats give best labeling reactivity in FACS to these cells.

The reactivity to the other recombinant (human/mouse) LRCC15 expression constructs is rather weak, please comment.

To confirm the identified markers, a histopathological study on thin sections with implanted U87-MG cells (glioblastoma model) in mice would be helpful for the validation. Indeed, this could also respond to the question of the accessibility of the targeted antigen. GFP expressing-U87-TG cell lines (largely available) could allow a quantitative evaluation of the labeling with the F4 and B3.1 in Cys-ScFv and Cys-Db formats, by histology.

Methods

The manuscript describes the development to obtain a labeling marker by the

phage-display ScFv libraries approach for the fast-track development of small antibodies.

Although the methods are well described, some informative details are not provided.

Some examples

- for immunization of mice, (for ethical concerns) the adjuvants should be named.

- the primers for PCR are not given; this is necessary for the reproducible data development.

Minor : name of immunized mice,  Balb/c (not balb/c)

Reviewer 2 Report

This manuscript presents development of the phage-display ScFv libraries that could be used for tracking/testing of anti-LRRC15 small fragments that could help further in establishing of quick diagnostics and prevention measures.

The manuscript is written in a consequential and organized mode. I would suggest adding some relevant references, especially for the methods/techniques used directly or modified in this study.

The study could be evaluated as comprehensive if we consider that the currently available and proposed validation system of used methods is complete.

The manuscript is one step forward to improve and enlarge the functional diversity of phage-display ScFv libraries for diagnostics. Thus, the manuscript will be interesting and useful for the readers of “International Journal of Molecular Sciences”, demonstrating them power of the discovery of high-affinity binders against various types of targets.

I would give some suggestions/comments to be addressed:

Line 23.  should be (LRRC15) instead of (LRCC15).

Line 28.  Show here what “h” stands for, like this Human LRRC15 (hLRRC15) as further you are using hLRRC15.

Line 29. Define abbreviation ScFv (single-chain variable fragment) as it appears first here.

Line 33. Suggest to write “that was further used in site-specific radiolabeling tests” instead of “Usable for further site-specific radiolabeling tests”.

Lines 34-35. “The library obtained was 1.2 x107 cfu”.

Define cfu (colony forming unit) here, or in the manuscript body.

And per what? Transformation efficiency …μg?

Line 36. Missing space “34 hLRRC15specific”.

Line 38. 1.0 to 4.5 mg instead of 1 to 4.5 mg.

Lines 127-133. Give reference to the method that was used as a basic or modified for immunization protocol and FC separately.

Line 132. Give definition how it was evaluated as “very good reproducibility”.

Line 146 and 151. Give reference to the methods that was used as a basic or modified for “PCR, gel-electrophoresis and purification”.

Line 151. Give short explanation of “proper”, I understand you mean size, but provide kind of standard for proving to be right (proper) one.

Line 157. It is better to write 3 rounds of PCR, everywhere.

Line 163. Give explanation why it was “an acceptable size was >106 cfu”? And it is better to write for the given testing conditions instead of “for us”.

Line 166. Again, better to write that IR was assessed as acceptable when it was higher than 80% of positive clones.

And give short explanation how it was assessed.

Lines 168-170.

-        Give definition to “phagemid”.

-        “Cloned” could be removed: “clone sequences in the phagemid developed by us.

-        Instead of Good, sugegst to write

The effectiveness of selected open reading frames was validated by the sequencing of 96 randomly chosen positive PCR clones, considering that the number of coding sequences has to be higher than 80% (give reference here). The global diversity is also appreciated by a low level of identical sequences (less than 10%) to validate the library (give reference here).

Or sentence needs rephrasing.

Line 172. Describe why it was evaluated as “very good results”, Like that high percentages of obtained results demonstrates the accuracy of the procedures/experiments performed.

Lines 173-174. In the Table 2: define pfu (plaque forming unit) and

Add per/what for both cfu and pfu

Line 176: suggest to write very high % instead of huge %

Line 178. suggest to write effective ORF

Line 183. “The selection results obtained were very satisfactory” – give the reference to compare or kind of standard to define/explain why it was satisfactory.

Lines 188-206. Give reference to the methods that was used as a basic or modified for…

Line. 217-219: reference

Line 225. the purity was high good (>85%)

Line 289. Suggest to write very high good insertion (>95%)….and large huge diversity.

Lines 290-291. rephrasing: the optimized protocol developed by us, allows to design a library ready to be used for…

Lines 339-340. rephrasing: produce different fragments of Cys antibody.

Line 276. suggest to write based on our results instead of ”in our case”.

Lines 360-367. Give reference to the method that was used as a basic or modified for 4.1.1

Lines 377-381. Give reference to the method that was used as a basic or modified for 4.2

Lines 401-405. Give reference to the method that was used as a basic or modified for 4.4

Lines 410-440. Give reference to the method that was used as a basic or modified for 4.5.1

Line 445. Suggest to remove times “10 to 106 times”.

Line 447. Suggest to write to validate the high effectiveness of insertion and reading frames instead of “to confirm the good insertion and reading”.

Line 446. Give short explanation why was it “expected” size for positive inserts was near 1 100 bp.

Lines 452-455. Give reference to the method that was used as a basic or modified for phage infection and propagation.

Line 460. Add the different otherwise it says that there was used something else beside the cells

“were performed on the different cells.”

And give reference to each method/step that was used as a basic or modified for 4.6.1

Lines 473-479. Give reference to the method that was used as a basic or modified for 4.6.2

Lines 505-511. Give reference to each method/step that was used as a basic or modified for…

And add discretion to the title Validation of the product or method.

Round 2

Reviewer 1 Report

By considering  my suggestions and comments (except some confidential information being part of protected industrial ownership and the suggested "histopathology to confirm the identified biomarker for targeted in vivo labeling of cells in cancer microenvironment" ) and  by including supplement information,  the revised version of the manuscript provides significant improvement.